# CogDevelop2K: Reversed Cognitive Development in Multimodal Large Language Models

## Abstract

Are Multi-modal Large Language Models (MLLMs) stochastic parrots? Do they genuinely understand? This paper aims to explore the core cognitive abilities that human intelligence builds upon to perceive, comprehend, and reason in MLLMs. To this end, we propose **CogDevelop2K**, a comprehensive benchmark that spans 12 sub-concepts from primitive knowledge like object permanence and boundary to more complex abilities like intentionality understanding, structured via the developmental trajectory of a human mind. We evaluate 46 MLLMs on our benchmarks. Surprisingly, we observe a **reversed** cognitive developmental trajectory compared to humans. Comprehensively, we further evaluate the influence of evaluation strategies and prompting techniques.

## 1 Introduction

Building on the foundation of advanced large language models (LLMs), multi-modal large language models (MLLMs) have recently demonstrated human-level performance in complex tasks involving high-level reasoning, perception, and cognition (Li et al., 2024a; Liu et al., 2024; Gemini, 2023; Fu et al., 2023; OpenAI, 2023), such as Spatial Reasoning (Chen et al., 2024; Cai et al., 2024), OCR (Mori et al., 1999), Scene Understanding (Cordts et al., 2016; Chen et al., 2017), Action Recognition (Jhuang et al., 2013; Herath et al., 2017) and Prediction (Lan et al., 2014; Kong & Fu, 2022). The progress in MLLMs has reignited hopes for achieving Artificial General Intelligence (AGI). However, recent studies have shown that even state-of-the-art MLLMs face critical limitations as compared to human intelligence. On the one hand, they perform poorly on rudimentary reasoning tasks like counting Paiss et al. (2023), compositional reasoning Yuksekgonul et al. (2022) and spatial reasoning Hoehing et al. (2023) despite their excellence at high-level reasoning tasks on similar domains (Paiss et al., 2023; Rahmanzadehgervi et al., 2024). On the other hand, said excellency often does not appear to translate to more generalized and real-world contexts (Shiffrin & Mitchell, 2023; Zhang et al., 2024). To explore the underlying reason for these limitations, we draw inspirations from human cognitive development.

Past research has shown that humans exhibit a series of rudimentary yet robust abilities in domains such as object, number, space, action, and social cognition at a very young age. Such abilities, often known as "core" cognition, grounds the set of diverse and complex abilities of human intelligence that develop later (Spelke et al., 1992; 1994; 1995; Spelke & Kinzler, 2007; Baillargeon & Carey, 2012; Mitchell, 2020; 2021). From infancy to early adulthood, human cognition develops along a structured trajectory, with interdependent relations between early, simple abilities and late, complex abilities. For instance, the ability to imagine the perspectives of others typically develops between the ages of 3 and 6 (Piaget & Inhelder, 1969), while the capacity to fully comprehend others' intentions matures around age 12 (Wimmer & Perner, 1983; Wellman et al., 2001; Liu et al., 2008). At the same time, the ability to understand other people's intentions largely depends on the the ability to understand other people's perspectives (Iacoboni, 2009; De Waal & Preston, 2017; Liu et al., 2017; Caviola et al., 2021; Ninomiya et al., 2020). An influential account of human learning has suggested that cognitive development is fundamentally driven by the increase of computational/representational power of the system, which allows for more complex mental operations to be performed on external data (Fodor, 1975; Pylyshyn, 1980; Halford et al., 1998; Fodor, 2008). However, while high-level abilities emerge directly due to enhanced operational resources, these operations are critically guided by the "core" cognition system that has enabled the system to possess a rudimentary understanding of each cognitive domain. This early-stage grounding not only empow-

ers humans to achieve a reliable performance at basic yet widely-applicable tasks starting from very young ages, but is also precisely what supports high-level abilities to robustly direct task-relevant behaviors despite the nuanced signals exist in the environment (Mitchell, 2021).

Given this line of evidence, we suggest that the absence of a core cognition system may provide a joint account of the two limitations faced by current MLLMs: poor performance on basic reasoning and the lack of robustness with high-level reasoning. In order to assess this hypothesis, We draw on theoretical and empirical approaches from developmental science to create benchmarks that evaluate simple and complex cognitive abilities in large vision-language models that are interrelated along the developmental trajectory. On a high level, we follow Jean Piaget's theory of cognitive development, which identifies four stages in children: sensorimotor, preoperational, concrete operational, and formal operational (Piaget, 1950; Piaget & Inhelder, 1969; 1974). During the sensorimotor stage, infants acquire knowledge through sensory experiences and actions, developing an understanding of basic object properties, such as permanence, continuity, and boundaries. In the preoperational stage, symbolic representation emerges, along with a grasp of basic physical properties. The concrete operational stage is characterized by the development of logical thinking and an understanding of intuitive physics. Finally, the formal operational stage introduces more advanced cognitive abilities, including abstraction, hypothetical reasoning, counterfactual thinking, and tool use. The interdependence and developmental trajectories of these abilities can be mapped in terms of a tree-like structure (as illustrated in Fig. 1).

To evaluate the performance of MLLMs on the core cognitive abilities, we curate the first-ever vision cognitive development benchmark, termed as CogDeveop2K, which consists of a total of 2519 questions with 2517 images and 455 videos. Then, we evaluate 46 MLLM models on our benchmark that spans all four stages of cognitive development. We introduce a novel multi-frame question format to evaluate models' co-reference, cognitive reasoning and temporal understanding capability simultaneously. Forty-seven models are compared against a human baseline under zero-shot conditions using 11 different prompts (including no prompt). Surprisingly, while prompts can boost model performance by 8.1%, models still demonstrate reversed trends in cognitive development against those observed in children.

## Human Cognitive Development

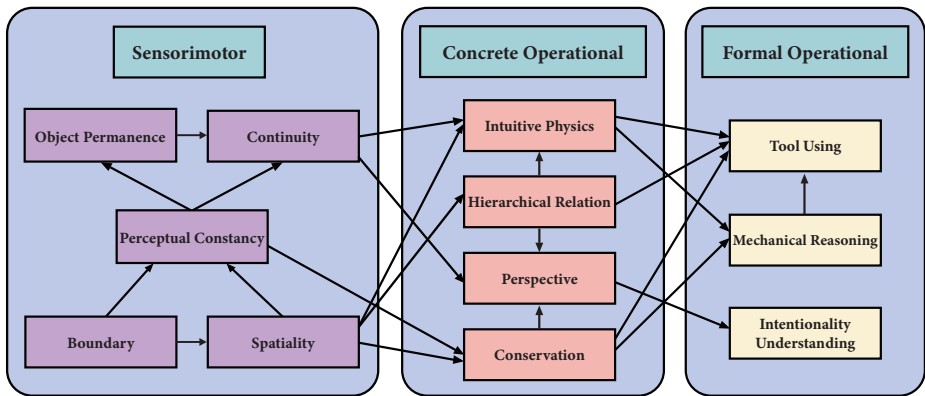

Figure 1: Map of core cognitive concepts during human developmental stages

## 2 COGDEVELOP2K

### 2.1 MULTI-MODAL LARGE LANGUAGE MODELS

The Vision Language Model (VLM) has a long history from Convolution Neural Networks (CNN) and Recurrent Neural Networks (RNN) (Karpathy & Fei-Fei, 2014; Vinyals et al., 2015) to unified modeling of visual and text modality with transformers (Li et al., 2019; Xu et al., 2023; Tan & Bansal, 2019; Alayrac et al., 2022; Radford et al., 2021). With the advancement of Large Language

Figure 2: A video-image interleaved example of multi-frame questions. To correctly infer the answer, model needs to understand the question by mapping each image (co-reference) to its option letter, to understand correlation between frames (temporal understanding) and to infer the possible trajectory of the bottle (reasoning).

Models (LLMs), existing state-of-the-art MLLMs (Liu et al., 2024; Li et al., 2023) adopt open-sourced Large Language Models such as Llama (Touvron et al., 2023), Mistral (Jiang et al., 2023). Instruction tuning is also introduced to further improve the task generalization ability of MLLMs (Liu et al., 2024; Dai et al., 2023). To acquire open-ended conversation abilities, LLaVA (Liu et al., 2024) proposes to distill the conversational abilities of ChatGPT to MLLMs, boosting performance by a large margin, which becomes a defacto procedure in the area (Wang et al., 2023; Bai et al., 2023; Gemini, 2023; Team, 2024; Sun et al., 2023; Li et al., 2022).

## 2.2 HUMAN COGNITIVE DEVELOPMENT

The sensorimotor stage is the first stage of cognitive development proposed by Jean Piaget (Piaget, 1952; Piaget & Inhelder, 1974). Spanning from birth to approximately 2 years of age, this stage is characterized by infants' understanding of the world through their sensory experiences and motor actions. Several prominent features of human intelligence develop during this period. First, infants develop object permanence, that they realize objects and people continue to exist even when not in direct sight, or being heard or touched (Baillargeon et al., 1985). They start to understand that there is a sense of continuity for the ways that objects exist, and the inductive bias of continuity is essential, e.g., for recognizing objects when occluded or for continuously tracking objects (Spelke et al., 1995; Le Poidevin, 2000). Infants also develop the sense of boundary during this stage, namely, the ability to recognize where one object ends and another begins (Kestenbaum et al., 1987; Jackendoff, 1991). Lastly, infants develop spatial and perceptual constancy by the end of sensorimotor stage. Spatiality refers to the ability to perceive the position and distance of objects relative to oneself and each other, and recognize the spatial invariance between them when presented by various sensory experiences (Hermer & Spelke, 1996; Bell & Adams, 1999).

Preoperational and concrete operational stage are the second and third stage of Piaget's cognitive development. Typically spanning over 2 to 7 years of age, preoperational stage is the transitional stage to concrete operational stage, which children enter around 7 years of age. During this period, children begin to develop internalized mental actions supported by organized structures that can be manipulated and reversed in systematic ways, known as mental operations (Janet, 1905; Kirkpatrick, 1908; Piaget, 1950; Piaget & Inhelder, 2014; Miller, 2016). Through mental operations, children are then able to rigidly perform tasks that are previously unreachable, such as thinking from other people's perspectives, understanding hierarchical relations of objects, and reasoning about physical events in the world. These tasks require not only rudimentary understandings of physical concepts, which gradually became in place during preoperational stage, but also relational and transformational reasoning that can only be done through mental operations (Piaget & Inhelder, 1974; Church & Goldin-Meadow, 1986; Houdé, 1997). Since preoperational stage is mostly meaningful as the transitional period preceding concrete operational stage, we do not have evaluation dimensions specifically targeting the stage. However, tasks targeting concrete operational stage could assess the existence of knowledge associated with preoperational stage, such as law of conservation (Piaget, 1952; Halford, 2011; Houdé, 1997).

The formal operational stage is the fourth and final stage in Piaget's theory of cognitive development, typically emerging around 11 or 12 years of age and continuing into adulthood (Inhelder & Piaget, 1958). Starting in this stage, one is able to systematic and flexibly apply mental operations to

not only concrete, physical domains but also abstract, formal domains (Kuhn & Angelev, 1976; Shayer, 1979; Huitt & Hummel, 2003). Foremost, this stage is characterized by the development of complex thinking and reasoning abilities, such as abstraction, pattern recognition, the employment of logic, and hypothetical and counterfactual reasoning (Piaget, 1950; Inhelder & Piaget, 1958). These cognitive advancements pave the way for more sophisticated abilities to interact with the physical world, marked by mechanical reasoning and tool use (O'Brien & Shapiro, 1968). Together, there is the advancement in social cognition, characterized by a deeper understanding of intentions, actions, and the reasoning behind them (Meltzoff, 1999).

### 2.3 Evaluation Dimension

**Boundary** Boundary refers to the cognitive understanding of where one object ends and another begins, an essential aspect of perceiving and understanding the physical world (Kestenbaum et al., 1987). Without understanding boundary, it seems very hard to construct a concept of object (Berkeley, 1709; Jackendoff, 1991).

**Spatiality** Spatiality, particularly demonstrated through the A-not-B task, involves a child's understanding of the location of objects in relation to their environment (Bell & Adams, 1999). In a classic A-not-B task, an object is hidden at location A (such as under a cup) and the child successfully finds it several times. Then, the object is visibly moved to a different location B (under a different cup), in full view of the child. Younger infants often make the error of searching for the object at the original location A, indicating a developmental stage where their understanding of object spatiality is still forming.

**Perceptual Constancy** Perceptual constancy is the cognitive ability to perceive objects as being constant in their properties, such as size, shape, and color, despite changes in perspective, distance, or lighting (Rutherford & Brainard, 2002; Khang & Zaidi, 2004; Green, 2023). For instance, consider a red ball being thrown in a park. To an observer, the ball appears smaller as it moves farther away, yet the observer understands it remains the same size throughout its trajectory.

**Object Permanence** Permanence, or specifically object permanence, is the idea that objects continue to exist even when they are not visible (Baillargeon, 1986; Spelke et al., 1992). Imagine a simple scene: a small child playing peek-a-boo. In the beginning, when the caregiver covers their face with their hands, the child might seem surprised or even distressed, thinking the person has disappeared. However, as children's understanding of permanence develops, they begin to realize that just because they can't see the person's face, it doesn't mean the person is gone.

**Continuity** Continuity is the cognitive prior in humans that in our world, objects usually exist in a consistent and continuous manner, even moving out of sight (Spelke et al., 1995; Le Poidevin, 2000; Spelke et al., 1994; Yantis, 1995; Yi et al., 2008; Bertenthal et al., 2013). Picture a train moving through a tunnel: as it enters one end, yet we naturally expect it to emerge from the other end, if the train is long enough. This expectation demonstrates our understanding of object continuity. Even though the train is not visible while it's inside the tunnel, we know it continues to exist.

**Conservation** Conservation refers to the ability to understand that certain properties of physical entities are conserved after an object undergoes physical transformation (Piaget & Inhelder, 1974). This is instantiated in their ability to tell that quantities of physical entities across different domains, such as number, length, solid quantity and liquid volume, will remain the same despite adjustments of their arrangement, positioning, shapes, and containers (Halford, 2011; Craig et al., 1973; Piaget & Inhelder, 1974; Houdé et al., 2011; Poirel et al., 2012; Marwaha et al., 2017; Viarouge et al., 2019). For example, when a child watches water being poured from a tall, narrow glass into a short, wide one, a grasp of liquid conservation would lead them to understand that the amount of water remains the same even though its appearance has changed.

**Perspective-taking** Perspective-taking is the ability to view things from another's perspective. This ability has seminal importance both to the understanding of the physical world as well as to the competence in social interactions (Wimmer & Perner, 1983; Wellman, 1992; Liu et al., 2008; Barnes-Holmes et al., 2004). The Three Mountain Task first invented by Jean Piaget is widely used

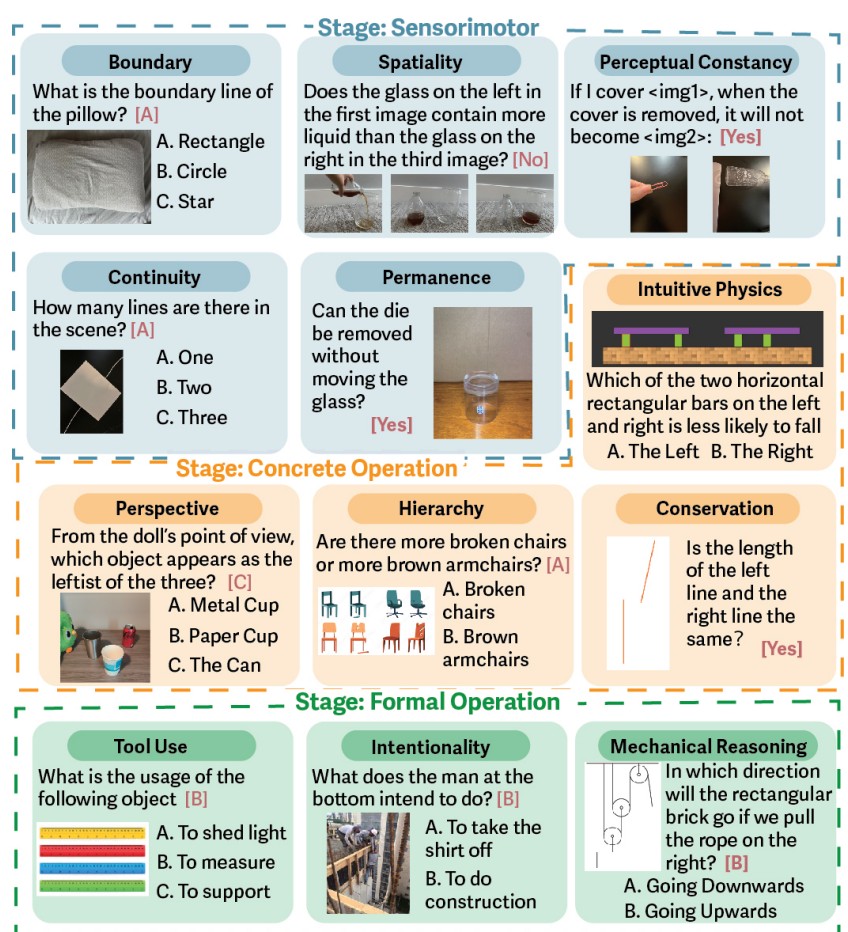

Figure 3: We demonstrate examples of different sub-concepts from the three stages.

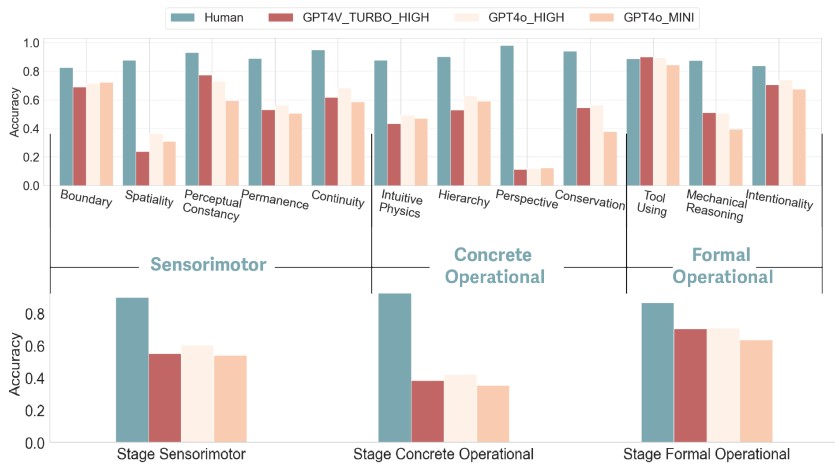

Figure 4: Reversed Cognitive Development in Advanced Models

in developmental psychology laboratories as the gold standard for testing perspective-taking abilities in children (Piaget & Inhelder, 1969)

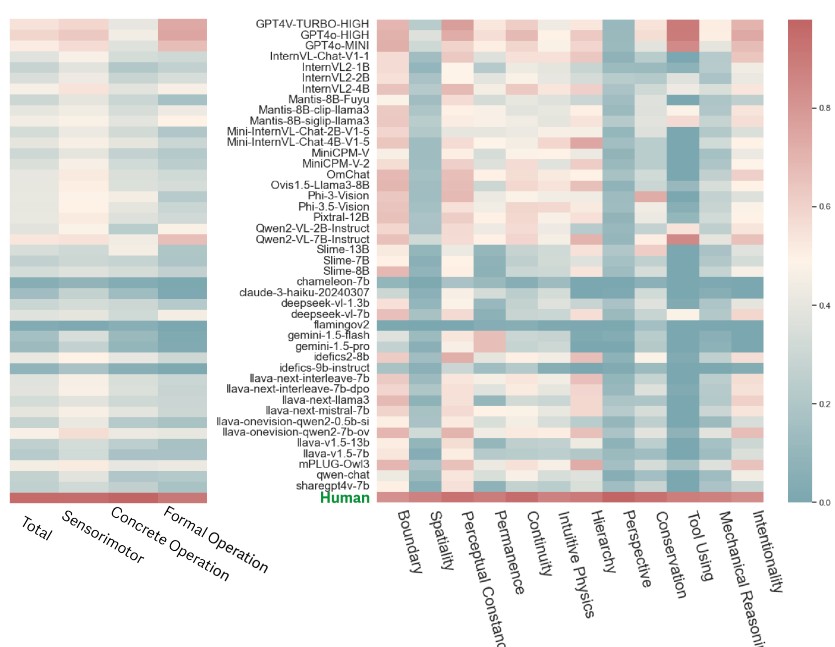

Figure 5: Multimodal Large Language Models and Human Performances

**Hierarchical Relation**   Hierarchical relation refers to the ability to organize objects or concepts into structured categories and subcategories, which are supported by the development of mental operations marked by class inclusion and transitivity (Shipley, 1979; Winer, 1980; Chapman & McBride, 1992). Class inclusion refers to the ability to recognize that some classes or groups of objects are subsets of a larger class. For example, a child in the concrete operational stage is able to understand that all roses are flowers, but not all flowers are roses (Borst et al., 2013; Politzer, 2016). This concept is essential for one's systematic and logical organizations of conceptual knowledge. Transitivity refers to the ability to understand logical sequences and relationships between objects (Andrews & Halford, 1998; Wright & Smailes, 2015). For instance, if a child knows that Stick A is longer than Stick B, and Stick B is longer than Stick C, they can deduce that Stick A is longer than Stick C.

**Intuitive Physics**   Intuitive physics refers to the ability of humans to predict, interact with, and make assumptions about the physical behavior of objects in their world (Michotte, 1963). As children grow, they transition from simplistic understandings, such as expecting unsupported objects to fall, to more complex theories, such as grasping the principles of inertia (Spelke et al., 1994; Kim & Spelke, 1999) and gravity (Vasta & Liben, 1996; Kim & Spelke, 1999; Li et al., 1999).

**Intention Understanding**   Intention understanding involves recognizing and interpreting the actions of others (Searle, 1979; Rosenthal, 1991). This process is not just about observing a behavior but also about understanding the goal behind it (Baker et al., 2009; Gandhi et al., 2021). For example, seeing someone reaching for a cup is not just about recognizing the physical action but understanding the intention behind it (e.g., they want to drink).

**Mechanical Reasoning**   Mechanical reasoning refers to the ability to understand and apply mechanical concepts and logical principles to solve problems (Allen et al., 2020). This cognitive concept first involves the ability to interpret and predict the behaviors of complex physical systems and understand how different mechanisms of the systems work. Second, mechanical reasoning requires the ability to apply logic rules, such as induction, abduction, syllogism (O'Brien & Shapiro, 1968; Cesana-Arlotti et al., 2018), and reasoning forms, such as hypotheticals and counterfactual (Byrne, 2016), to figure out how to manipulate these systems to achieve a desired outcome (Hegarty, 2004).

Table 1: Main statistics in CogDevelop2k. All the questions are image/video-text interleaved.

| Statistic | Number |
|---|---|
| single-frame | 1677 |
| multi-frame | 842 |
|     * multiple images | 200 |
|     * single video | 401 |
|     * multiple videos | 124 |
|     * video-image-text | 117 |
| total | 2519 |

**Tool Using**    Tool using refers to the ability to manipulate tool objects in its environment as aids in achieving a specific goal, such as obtaining food or modifying the surroundings. A lot of cognitive components involved in tool using ability, such as affordances, referring to computing the action possibilities offered to the agent by the tool with reference to the agent's sensorimotor capabilities (Gibson, 1979). For example, a door handle affords pulling or pushing, as how the door should be operated by a human agent.

## 2.4 DATA SOURCE

CogDevelop2K comprises 2517 images and 445 videos with multimodal options and questions, crawled primarily from networks as well as self-recorded content. Our targeted platforms include Wikipedia, Reddit, Twitter, Quora, and TieBa. Some of the captions were adapted from user comments to ensure content diversity and relevance. Videos for intuitive physics were either self-recorded or produced using Physion[1].

All concept questions were annotated by four researchers with cognitive science and computer science background, then reviewed by two independent researchers. For a question to pass the screening stage, a minimum correctness rate of $95\%$ was required from both reviewers.

## 2.5 DATASET DESIGN

Existing datasets typically support only one question-answer format or single modality type, which hinders the assessment of reasoning capabilities across different modalities within the same domain. For instance, current interleaved image understanding and video understanding models cannot be effectively compared on the same question. To address this limitation, we include multiple Q&A formats (e.g., multiple-choice, true/false, and numeric question-answer) and complicate question-answering by incorporating a new image-video-text interleave format as shown in Fig. 2. To further explore the cognitive development capabilities of models across these modalities, we optimize our formality as follows:

**Addressing Weak Image-Text Correlation and Imbalance**    In existing interleaved image-text datasets, the correspondence between images and text is often loose, and text provides marginal information for image modeling. This imbalance can cause models to over-rely on textual information, especially when text segments are lengthy (Lin et al., 2023) . To address this issue and focus on the image understanding abilities of the model, we eliminate sentences that describe the image. This ensures that the textual information is highly relevant to but does not overlap with the image content.

**Testing Co-Reference, Reasoning, and Temporal Understanding with novel Multi-Frame Questions**    Multi-frame questions can simultaneously evaluate a model's three inference ability: *Co-Reference*, *Reasoning*, and *Temporal Understanding* (Jiang et al., 2024). Co-reference involves linking natural language descriptions with specific image inputs (e.g., "the first image" or "A."). Reasoning requires models to make decisions based on cognitive knowledge, such as describing spatial relationships. Temporal Understanding, on the other hand, tests the model's capability to

---

[1] https://physion.net/

Figure 6: MLLMs' Dissociation Between Law of Conservation and Rudimentary Quantity Understanding as Exemplified by GPT-4o

comprehend sequences of frames in terms of temporal order (multi-frame) and correlation (multi-view) (Li et al., 2024b). Existing interleaved multi-image datasets can not adequately test all three properties simultaneously. For example, video datasets with temporal information often include only a single video, while multi-image datasets that require co-reference lack temporal dependencies. To address this, we introduce multi-video interleaving and video-image interleave formats (multi-frame) to evaluate all three properties concurrently. The statistics of the dataset are presented in Table 1.

## 2.6 EVALUATION STRATEGY

We comprehensively evaluate models' capability of cognitive reasoning using *46 multi-image interleave MLLMs* with 11 different promopts. The two evaluation baselines are outlined as follows:

**Human baseline** We recruit 22 participants, all of whom are college students proficient in English. Participants are instructed to skip a question if the question is phrased ambiguously or is too complicated to answer in 90 seconds. This question is marked as failed if the human participant does not provide an answer.

**Zero-Shot-$448^2$-Circular Baseline** Similar to previous studies (Lu et al., 2022), the zero-shot setup follows the format of $Q(M)T \rightarrow A$, where the input includes the question text (Q), task description (T), and multiple options (M) concatenated as tokens, with the output being the predicted answer (A). Given that model predictions can exhibit bias in multiple-choice settings, we implemented circular evaluation as baseline. In circular evaluation, all answer options are shifted one position at a time, ensuring that the correct answer appears in each option slot. Only when the model correctly predicts all shifted answers is it considered accurate (Liu et al., 2023). All images and videos were resized to $488^2$.

**Prompts** Strategically crafted prompts can enhance model performance, regardless of whether fine-tuning is applied (Bsharat et al., 2023; Yang et al., 2023). To mitigate this, we use image-independent contexts, such as relevant concept introductions and character assignments, which encourage models to reason beyond the provided textual information. The prompts we used can be categorized into leading words, deeper thing, role assignment, reward or penalty, and explanation.

## 3 RESULTS

We systematically evaluate 48 Multi-modal Large Language Models on the CogDevelop2K benchmark, which spans 12 cognitive concepts designed to assess a broad range of the developmental trajectory of Multi-modal Large Language Models. These abilities substantiate core cognition ranging from object permanence and boundary to mechanical reasoning and intentionality understanding. The models were tested across multiple question formats and ten prompt variations, yielding a comprehensive assessment of their core cognition. For example, in the sensorimotor stage, GPT families show moderate performance, with accuracy scores between 0.4 and 0.6. In the concrete operational stage, GPT families show lower performance, with accuracy scores between 0.2 and 0.4. Nevertheless, in the formal operational stage, GPT families show stronger performance, with accuracy scores

Table 2: Evaluation of different prompting techniques. The best result is achieved when the concept explanation is provided to the model. We highlight the improvement over empty string in red.

| Category | Prompt | GPT4V Turbo High | GPT4o High | GPT4o Mini |
|---|---|---|---|---|
| | Empty String | 0.519 | 0.555 | 0.487 |
| **Leading** **Word** | 1. Let's think step by step. | 0.531 | 0.577 | 0.489 |
| | 2. Take a deep breath and answer this question carefully. | 0.522 | 0.562 | 0.489 |
| **Deeper** **Thinking** | 3. Please answer the question and provide an explanation. | 0.518 | 0.562 | 0.499 |
| | 4.Please answer the question and explain to me in simple terms. | 0.476 | 0.569 | 0.501 |
| | 5. Please answer the question and ensure that your answer is unbiased and doesn't rely on stereotypes. | 0.522 | 0.575 | 0.478 |
| **Role** **Assignment** | 6. (Assign assistance's role) You are an expert on cognitive science and are familiar with [Concept name] | 0.565 | 0.617 | 0.545 |
| | 7. (Assign audance's role) Please answer the question and explain it to me like I am 11 years old. | 0.538 | 0.564 | 0.496 |
| **Reward** **&** **Penalty** | 8. Please answer the question carefully. I'm going to tip you 200 dollars for a better solution. | 0.528 | 0.563 | 0.487 |
| | 9. Please answer the question carefully. You will be penalized if your answer is incorrect. | 0.522 | 0.566 | 0.491 |
| **Explanation** | 10. Please read the concept explanation and then answer the related question. Concept: [concept description]. | **0.586** (+ 0.067) | **0.636** (+ 0.081) | **0.547** (+ 0.06) |

between 0.6 and 0.8. Surprisingly, we find an inverse cognitive developmental trajectory compared to humans in more advanced models, which are typically regarded as state-of-the-art (Fig. 3 and Fig. 4).

**Influence of Prompts.** We investigate the influence of different prompting techniques on the performance of MLLMs on our benchmark. As illustrated in Table 2, we explore 10 different prompting techniques (divided into 5 categories). We observe that most prompts are useful on our benchmark, increasing the averaged performance by at least 1%. Concept explanation, which offers a clearer context of the question to the MLLMs, surpasses all the other prompts by at least 6%.

## 4 DISCUSSION

Our results demonstrate that MLLMs exhibit an intriguing reverse cognitive development pattern. Namely, they are systematically proficient at complex tasks that are typically understood to require abilities underlying simple tasks that they perform poorly. This finding supports the hypothesis that MLLMs lack a "core" cognition system, which, as we suggested, could appear as challenge to the current foundational architecture of MLLMs as a long-term solution to achieve human-like general intelligence (Summerfield, 2022). Specifically, the inability to implement core knowledge in artificial intelligence models prevent them from achieving human-level robustness in performances, even if such models seem to excel at certain complex cognitive reasoning tasks (Mitchell, 2020; 2021; Shiffrin & Mitchell, 2023; Palmarini & Mitchell, 2024). MLLMs' poor performances on foundational concepts like spatiality, permanence, continuity, and perspective-taking, which directly reflect

upon grasps of core knowledge, while achieve proficiency in complex concepts like tool using and intention understanding exactly exemplifies this concern.Specifically, the developmental trajectory of human cognition is marked by complex cognitive abilities being grounded upon extremely robust understandings of a series of foundational concepts, namely core knowledge (Spelke, 2000; Spelke & Kinzler, 2007). Through early stages of development, children exhibit rudimentary yet stable understandings of objects, actions, number, space, and social partners, each dimension laying the foundations for the acquisitions of complex abilities in later life.

While a straightforward explanation of this intriguing reverse cognitive development pattern is the absence of "core" cognitive abilities, further research is needed in order to arrive at a comprehensive explanation for the mechanistic details of why such abilities fails to emerge as opposed to the case in humans. At the same time, existing theories may provide some preliminary insights into this question. In particular, it is likely that scaling of parameters and training data mainly enhanced the operational resources available to MLLMs but not necessarily allow the grasp of core knowledge (Fodor & Pylyshyn, 1988; Kello et al., 2010; von der Malsburg, 2024). This is because neural networks do not themselves contain built-in domain-specific information but are only capable of representing such knowledge through the topological structures of the connection units and the values of their corresponding weights. Therefore, while MLLMs may have gathered all the information needed for constructing a system of core knowledge from the training data , it is likely represented in an extremely distributed fashion that prevents reliable tracing and retrieval (Hinton et al., 1986; Fodor & Pylyshyn, 1988; Yang et al., 2022). If the task condition do not offer enough cues to trigger the required aspect of the system, then MLLMs may fail at the task even if they "know" the concept that is needed for providing the correct answer.

An initial validation of this explanation is that the best model performance on our benchmarks are achieved when the prompts contain an explanation of the concept tested by the experiment (see Table 2). Such an effect is notable given that unlike regular in-context learning prompts, concept explanations contain mainly high-level, abstract information that has no direct relevance to the context of the task. Oppositely, it may be seen as a rather immediate attempt of conceptual pre-training that facilitates the retrieval and synthesis of "core" conceptual resources distributed throughout the networks. The effectiveness of such an attempt calls for the attention to training measures for MLLMs aiming at bettering the implementation and use of core cognition, which appears to be a promising approach toward more robust and comprehensive foundational models.

## 5 CONCLUSION

In this paper, we explored the cognitive capabilities of Multi-modal Large Language Models (MLLMs) through the lens of core cognitive abilities that underpin human intelligence. By introducing CogDevelop2K, a novel benchmark that spans 12 subconcepts across developmental stages, we aimed to assess the fundamental understanding and reasoning capacities of MLLMs. Our evaluation of 46 models revealed intriguing insights, including a reversed cognitive developmental trajectory compared to humans. This finding raises questions about whether MLLMs truly comprehend tasks or simply exhibit performance without genuine understanding. These results underscore the need for further investigation into the cognitive foundations of MLLMs, as well as the influence of evaluation strategies and prompting techniques in shaping their outcomes. Ultimately, this study serves as a step toward unraveling the nature of MLLM intelligence and their potential limitations in mirroring human cognition.

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
