# OpenReview forum: "CogDevelop2K: Reversed Cognitive Development in Multi-modal Large Language Models"
_ICLR.cc/2025/Conference — Submitted to ICLR 2025_

### Official Review · Reviewer_7m3D · 2024-10-27

**Soundness:** 3
**Presentation:** 3
**Contribution:** 3
**Rating:** 6
**Confidence:** 4

**Summary:**

The paper investigates whether Multi-modal Large Language Models (MLLMs) truly understand the tasks they excel at or if they are simply mimicking intelligence by learning spurious correlations. The authors aim to answer whether MLLMs possess genuine cognitive capabilities akin to human intelligence or if they are “stochastic parrots” that predict outcomes based on statistical patterns rather than comprehension. To explore this, the authors propose a new benchmark called CogDevelop2K, inspired by human cognitive development. CogDevelop2K comprises 2,519 questions paired with 2,517 images and 455 videos. The authors used different prompting techniques to assess a large set of MLLM models under different conditions. For the final conclusion,  the authors observed a reversed cognitive
developmental trajectory compared to human beings.

**Strengths:**

- The paper tackles a pressing question in the rapid development of the MLLMs. From my point of view, CogDevelop2K is a novel and well-designed benchmark that comprehensively spans multiple cognitive domains. It includes a wide variety of tasks, from simple core concepts like object permanence to complex reasoning, making it a robust tool for evaluating the cognitive capabilities of MLLMs. Human cognitive development theories are well-studied, while MLLMs are not. By linking AI performance to human developmental stages, the study provides a unique perspective and a structured framework for evaluation.
- The experiments are well-designed and comprehensive, and the dataset is extensive. I think the paper provides a generally broad and varied evaluation of currently popular MLLM models, and offering a thorough comparison of existing models’ strengths and weaknesses across cognitive domains.

**Weaknesses:**

- The study primarily focuses on the performance of MLLMs, which are heavily reliant on vast amounts of training data. This reliance raises questions about the models' true cognitive capabilities, as the impressive results may simply reflect the capacity to identify statistical correlations in massive datasets rather than genuine comprehension or understanding. It's difficult to determine if their cognitive abilities are a result of learning core principles or merely memorizing patterns. This limitation suggests that the current approach might not effectively distinguish between true intelligence and advanced pattern recognition. While I ack that the CogDevelop2K benchmark is an interesting contribution, the paper may not push the community forward in developing more sophisticated or truly intelligent AI systems from a technical view.
- Lack technical contribution. The study lacks a detailed technical analysis, making its findings primarily observational. This limits its impact on advancing MLLM development, as it does not offer a clear path for addressing the identified limitations or enhancing the cognitive capabilities of AI systems. While the paper provides valuable insights into the current performance of models, it falls short of proposing technical insights that could drive meaningful progress in the field.
- Some typos, i.e., line 66 "CogDeveop2K".

**Questions:**

- What factors do you believe contribute to the reversed cognitive developmental trajectory observed in MLLMs? Is it a consequence of the models' training data, architecture, or inherent biases in the dataset?
- How well do you think the CogDevelop2K benchmark generalizes to the applications or the development of MLLMs? Are there specific use cases where you believe MLLMs' performance on this benchmark accurately reflects their capabilities? Based on the evaluation results, how the CogDevelop2K can help improve the development of MLLMs?

---

> ### Author Response · Authors · 2024-11-28
> **Response to reviewer 7m3D (Part 1)**
>
> We thank the reviewer for recognizing our benchmarks' diversity and novelty and our evaluation's comprehensiveness. We also express our sincere gratitude for the valuable comments and suggestions, which we will address as follows:
>
> > W1: The study primarily focuses on the performance of MLLMs, which are heavily reliant on vast amounts of training data. This reliance raises questions about the models' true cognitive capabilities, as the impressive results may simply reflect the capacity to identify statistical correlations in massive datasets rather than genuine comprehension or understanding. It's difficult to determine if their cognitive abilities are a result of learning core principles or merely memorizing patterns. This limitation suggests that the current approach might not effectively distinguish between true intelligence and advanced pattern recognition. While I ack that the CogDevelop2K benchmark is an interesting contribution, the paper may not push the community forward in developing more sophisticated or truly intelligent AI systems from a technical view.
>
> Thanks for the comment. We acknowledge the difficulty in distinguishing between true cognitive capabilities and advanced pattern recognition. However, we do believe that a key contribution of our work is precisely to resolve the same issue the reviewer raises.. In particular, the finding that MLLMs excel at complex tasks but fail at simple tasks on the same domain suggests that their performance on these complex reasoning tasks do not reflect true understanding but are likely the product of exploiting spurious correlations. Simply, it is difficult to imagine someone excelling in complex tasks (e.g. solving advanced mathematical problems) to fail at rudimentary tasks on the same domain (e.g. counting) provided that they actually understand the problems. On a more technical basis, we further suggest that this is because core cognition emerges in humans at a very young age and supports the acquisition of further, high-level abilities, while such a system is lacking in MLLMs. By systematically examining the primitive core cognitive abilities, our work provides a measurement of the "true" capabilities of models.
>
> ---
>
> > W2: Lack technical contribution. The study lacks a detailed technical analysis, making its findings primarily observational. This limits its impact on advancing MLLM development, as it does not offer a clear path for addressing the identified limitations or enhancing the cognitive capabilities of AI systems. While the paper provides valuable insights into the current performance of models, it falls short of proposing technical insights that could drive meaningful progress in the field.
>
> We have updated the relevant technical analysis on the revision, which provides a preliminary account of the reason underlying the reversed developmental trajectory we observed and how it implies practical guidance for AI development. In short, we stress that the phenomenon is due to the inability to employ core knowledge in problem-solving. This suggests that further development of MLLM reasoning may benefit from pre-training the correspondence between the conditions of the task and the cognitive domain it is grounded upon, as highlighted by our data showing a unique improvement in performance caused by adding concept explanations in the prompts.
>
> ---
>
> > W3: Some typos, i.e., line 66 "CogDeveop2K".
>
> We have revised the manuscript and fixed all typos.
>
> Due to the enforced word limit, please see our responses toward the questions in the comment below.

---

> ### Author Response · Authors · 2024-11-28
> **Response to reviewer 7m3D (Part 2)**
>
> > Q1: What factors do you believe contribute to the reversed cognitive developmental trajectory observed in MLLMs? Is it a consequence of the models' training data, architecture, or inherent biases in the dataset?
>
> We believe this reversed cognitive development is caused by the foundational architecture of the  models and the training data. To begin with, while the scaling of parameters enhances the representational power of the system, they do not explicitly contain domain-specific information. Even if the core knowledge set  needed for solving these rudimentary tasks they failed at are gathered throughout the training process, it is likely represented distributedly across the entire system and is hard to be efficiently retrieved and applied during problem-solving, especially  If the task condition do not offer enough cues for the concepts needed in the task. This is again supported by our preliminary results showing that improvements in performance can be achieved simply by explaining the concepts in the prompts. At the same time, as other reviewers have also noted, the training data for MLLMs likely contains more information that is directly relevant to the task conditions assessing high-level abilities, which offers more spurious correlations to be exploited when solving the tasks.
>
> ---
>
> > Q2: How well do you think the CogDevelop2K benchmark generalizes to the applications or the development of MLLMs? Are there specific use cases where you believe MLLMs' performance on this benchmark accurately reflects their capabilities? Based on the evaluation results, how the CogDevelop2K can help improve the development of MLLMs?
>
> We think that by using specially curated dataset based on classic cognitive experiments, the CogDevelop2K benchmark in general measures the capabilities of MLLMs on the concepts it assessed in a fairly accurate way. In particular, given that the probing of simpler, basic concepts contain less contextual clues that are directly manifested in the training data, it should be predicted that their poor performance on these concepts are more likely to reveal true limitations in cognitive abilities, whereas their proficiency on high-level concepts are more likely to be contributed by advanced pattern recognition. We believe that the benchmark provides important insights toward the further development of MLLMs, particularly in that it reveals the current lack of core cognitive abilities in MLLMs and that it may underlie the limitations faced by MLLMs on high-level tasks regarding robustness and real-world generalizability. The benchmark can help improve the development of MLLMs by highlighting the pre-training of core concepts as a promising pathway and providing a large and relevant dataset based on the cognitive science literature to support such endeavors.

---

### Official Review · Reviewer_6bzH · 2024-10-27

**Soundness:** 2
**Presentation:** 2
**Contribution:** 2
**Rating:** 5
**Confidence:** 4

**Summary:**

The paper presents a new benchmark called CogDevelop2K, which presents visual reasoning questions that correspond to what the authors identify as different stages of human cognitive development.  The authors test both humans and multimodal LLMs on these questions and find that multimodal LLMs are better on questions corresponding to later stages of human cognitive development than they are on questions corresponding to earlier stages.

**Strengths:**

The paper divides evaluation into several dimensions that correspond to concepts associated with different developmental stages.  These include spatial reasoning, perceptual constancy, object permanence, temporal understanding and many others.  The dataset includes questions that use both single image frames and videos.  It is interesting, though not surprising, to see that Moravec's famous paradox holds here -- the tasks that are easier for LLMs are the ones that humans only accomplish at later developmental stages.

**Weaknesses:**

The paper's description of the developmental psychology literature seems outdated.  Most of the references to this literature are decades old, and there are few more recent references.  I'm not an expert on developmental psych, but I do know that there has been a lot of research in recent years that calls into question some earlier "accepted" results of, say, Piaget and others, even those working in the 1980s and 1990s.  The authors should cite more recent work here, and perhaps revise some of their claims about developmental stages accordingly.

The paper's main claim is stated in a way that I think is misleading: "we find an inverse cognitive developmental trajectory compared to humans".  This implies that LLMs "develop" along a "trajectory" that is the reverse of humans.  But of course LLMs do not develop along a trajectory (unless one wants to look at scaling trends or at models at different training checkpoints).  What's shown here is not a developmental trajectory, but rather a difference in what LLMs and humans find difficult, at different points in development for *humans*.  This should be made clearer as the central claim.  And perhaps reference Moravec's paradox.

The paper lacks some important references.  The authors state "we curate the first-ever vision cognitive development benchmark".  Other researchers have created AI benchmarks based on developmental psychology -- see e.g., https://arxiv.org/html/2406.10215v1 -- and on capacities that are part of the CogDevelop2K benchmark, e.g., physical reasoning (e.g., https://arxiv.org/pdf/2402.06119) and others.   The authors should be clearer on what their benchmark contributes, beyond the many "visual commonsense" benchmarks (e.g., https://visualcommonsense.com/) Also, there are many papers that investigate limitations of MMLMs on visual reasoning, and some of these should be cited.

There are many things in the paper that were unclear, which I ask about in the Questions section below.  There are also numerous grammatical errors and typos -- the paper should be carefully proofread.

**Questions:**

How were the questions for the benchmark created?  Did the authors come up with questions related to the different evaluation dimensiona and then search for images / videos to use for those questions?  Or was it some other process?

Will there be a link in the paper for downloading the benchmark?

The paper states: "For a question to pass the screening stage, a minimum correctness of 95% was required from both reviewers."  I didn't understand this -- what does "95% correctness" mean for a single question?

The paper states: "current interleaved image understanding and video understanding models cannot be effectively compared on the same question."  What does "interleaved image understanding and video understanding" mean?  [I don't know what you mean by "interleaved" here.]  And why can't current models be compared on the same question?

I also am not sure what you mean by "interleaved image-text datasets" and "multi-video interleaving and video-image interleave formats".

I really did not understand the description of the "Human Benchmark".  You say that there were 22 participants.  Then you say "Each annotator was asked to label 2 to 6 concepts".  Are the 22 participants acting as labelers, or as participants answering questions from the dataset?  What are the human labels used for?  You also say "Participants were instructed to skip a question if the question is worded ambiguously or is too complicated to answer in 90 seconds".  Are these participants answering questions from the benchmark?  What happened to those questions that people skipped?  Were they taken out of the benchmark and not used for LLM evaluation?    All this needs much clearer explanation.

Also, human performance is given in Figure 5.  How many humans were tested on each benchmark question?  How many benchmark questions were given to each human?  Say more about the human study -- how were they presented with the questions ? Were there questions given to LLMs that weren't given to humans?  Can you give some statistical significance measures to the differences between models and between humans and LLMs?

Also, Figure 5 is very hard to read.  It would be useful if you just picked the five best models, and gave their accuracies compared to humans in a table, and put the rest of the results in an appendix.  Also, for the LLM results, which prompt format was being used?

The paper states that for the questions in the benchmark: "the textual information is highly relevant to but does not overlap the image content".  It would help to give an example, or point to one already in paper.

The paper states: "GPT families show moderate performance, with accuracy scores between 0.4 and 0.6".  For these and other results, what is random chance baseline? Why do you call this "moderate" performance?

For Table 2,  What is the prompt in "empty string" condition  Just the question itself plus images?   Give an example of an actual prompt, including the question itself.  For all the prompt formats listed in Table 2, where does the prompt text go with respect to the question itself?

Also for Table 2: Are the values given in the table the accuracies over all questions in the dataset?

Table 2: "Concept: [concept description]".  Give an example of a concept description that would appear here.

"Concept explanation...surpasses all the other prompts by at least 6%". I'm confused by this.  GPT4o-High for example gets 0.636 with "explanation" prompt and "0.617 with "role assignment prompt."  This
is less than 3 percent difference.  There are other even closer examples in the table.

The paper states, for Table 2: "We observe that most prompts are useful on our benchmark, increasing
the averaged performance by at least 1%." Increased over what?

"MMLMs virtually do not understand the answers they produce when tackling complex reasoning tasks".  On this you should cite https://arxiv.org/abs/2311.00059

"This study serves as a step toward unraveling the nature of MLMM intelligence and their potential limitations in mirroring human cognitive development."  But why should mirroring human cognitive
development be a goal?

---

> ### Author Response · Authors · 2024-12-02
> **Response to reviewer 6bzH**
>
> > W1: The paper's description of the developmental psychology literature seems outdated. Most of the references to this literature are decades old, and there are few more recent references. I'm not an expert on developmental psych, but I do know that there has been a lot of research in recent years that calls into question some earlier "accepted" results of, say, Piaget and others, even those working in the 1980s and 1990s. The authors should cite more recent work here, and perhaps revise some of their claims about developmental stages accordingly.
>
> We appreciate the suggestion to discuss more recent works in developmental psychology. On the other hand, we would like to note that while a lot of recent research has questioned the early works of Piaget’s and others, these updates mostly target the empirical predictions regarding specific developmental parameters (e.g. the age when children tend to acquire object permanence or egocentrism), whereas Piaget’s framework emphasizing a stage-to-stage transition from simple to complex abilities remains to be supported, which is what we adopted here. At the same time, given that our discussions featuring the developmental literature focus mainly on how specific cognitive abilities are assessed in the benchmark (see particularly Section 2.3. Evaluation Dimension), we aim to cite studies that provide foundational insights regarding the effectiveness of the measures. In this case, whether more recent studies are included mostly depends on whether they appear to be critical in developing or advancing said measures.
>
> ---
>
> > W2: The paper's main claim is stated in a way that I think is misleading: "we find an inverse cognitive developmental trajectory compared to humans". This implies that LLMs "develop" along a "trajectory" that is the reverse of humans. But of course LLMs do not develop along a trajectory (unless one wants to look at scaling trends or at models at different training checkpoints). What's shown here is not a developmental trajectory, but rather a difference in what LLMs and humans find difficult, at different points in development for humans. This should be made clearer as the central claim. And perhaps reference Moravec's paradox.
>
> Thank you for this informative suggestion. We accept that“cognitive developmental trajectory” is subjected to different interpretations, but we believe this framing does have certain merits that cannot be covered by the difference in what LLMs and humans find difficult. Specifically, theories regarding the mechanisms underlying cognitive development generally support that certain high-level abilities we assessed here are in nature grounded upon simpler abilities on respective domains. Given such interdependencies between abilities, a trajectory highlighted by stage-to-stage transitions can be proposed. Stage models like Piaget’s can be used to assess an individual's developmental status by examining their performance on a series of gradually more complex abilities and to see where the results place them on the proposed trajectory. With the same logic, we could also observe the developmental status of MLLMs. It is just that their performance does not fit with the proposed trajectory established through decades of human research but instead seems to fit with a new trajectory. To this end, we use the term “developmental trajectory” based on how individuals are placed on the stage models and the interdependencies between abilities rather than attempting to uncover how abilities emerge along with scaling trends or training checkpoints.

---

> > ### Comment · Reviewer_6bzH · 2024-12-02
> > **Reply to W1 and W2 responses**
> >
> > W1: There have been many updates to the notion of Piaget's notion of developmental stages. See, e.g., https://journals.sagepub.com/doi/full/10.1177/17456916231186611 for an overview.  The paper should discuss and cite this literature.
> >
> > W2:  A "trajectory" implies a dynamic set of transitions over time.  What you are measuring in LLMs is not any kind of trajectory, so I recommend using a different term. Perhaps "hierarchy of capabilities", which in humans is developed over time.

---

> ### Author Response · Authors · 2024-12-02
> **Response to reviewer 6bzH**
>
> > W3: The paper lacks some important references. The authors state "we curate the first-ever vision cognitive development benchmark". Other researchers have created AI benchmarks based on developmental psychology -- see e.g., https://arxiv.org/html/2406.10215v1 -- and on capacities that are part of the CogDevelop2K benchmark, e.g., physical reasoning (e.g., https://arxiv.org/pdf/2402.06119) and others. The authors should be clearer on what their benchmark contributes, beyond the many "visual commonsense" benchmarks (e.g., https://visualcommonsense.com/) Also, there are many papers that investigate the limitations of MMLMs on visual reasoning, and some of these should be cited.
>
> Thanks for bringing our attention to these works. While we acknowledge other benchmarks inspired by cognitive to evaluate LLMs or MLLMs, we are the first comprehensive vision benchmark on the cognitive development for MLLMs with an emphasis on core cognitive concepts and lower-level cognitive abilities. The major difference between previous work like DevBench [1], ContPhy [2] and visual commonsense [3] is that they focus on part of the cognitive developmental process (such as linguistic in [1]) or higher-level abilities (e.g. physical reasoning [2] or visual commonsense reasoning [3]). While these works do provide key insights into the cognitive aspects of MLLMs and LLMs, they only focus on higher-level cognitive concepts and do not aim to provide a comprehensive analysis of the correlations between core cognition and high-level abilities. By analyzing this, we found that MLLMs lack simple abilities that provide groundings for high-level reasoning abilities in humans, supporting the concern that MLLMs’ current divergence from human-level robustness and real-world generalizability may be due to the lack of reasoning foundations provided by core cognition [4]. Please also see the discussion section of our updated paper for further analysis.
>
> [1] Tan, Alvin, Wei Ming, et al. "DevBench: A multimodal developmental benchmark for language learning." arXiv preprint arXiv:2406.10215. 2024.
>
> [2] Zheng, Zhicheng, et al. "ContPhy: Continuum Physical Concept Learning and Reasoning from Videos." arXiv preprint arXiv:2402.06119 .2024.
>
> [3] Zellers, Rowan, et al. "From recognition to cognition: Visual commonsense reasoning." Proceedings of the IEEE/CVF conference on computer vision and pattern recognition. 2019.
>
> [4] Shiffrin, Richard, & Mitchell, Melanie. “Probing the psychology of AI models.” Proceedings of the National Academy of Sciences, 120(10), e2300963120. 2023.
>
> ---
>
> > W4: There are many things in the paper that were unclear, which I ask about in the Questions section below. There are also numerous grammatical errors and typos -- the paper should be carefully proofread.
>
> We thank the reviewer for the suggestions. We have carefully proofread the paper and provide a revised version. We will continue to proofread and make our paper more comprehensive and clear. All rebuttal answers will be added to the final version of the paper.

---

> ### Author Response · Authors · 2024-12-02
> **Response to reviewer 6bzH**
>
> > Q1: How were the questions for the benchmark created? Did the authors come up with questions related to the different evaluation dimensiona and then search for images / videos to use for those questions? Or was it some other process?
>
> The benchmark’s overall framework was first structured through cognitive development literature. Question prototypes (e.g. three-mountain questions) are created through extensive literature surveys on old cognitive science papers on each concept revulsion dimension. Images and videos are curated through annotators’ efforts i.e. creating the cognitive science experiment instruments or attained from the internet. Then for each Image and video, annotators curated questions and ground-truth answers. All annotations have experienced undergraduate-level cognitive science training.
>
> ---
>
> > Q2: Will there be a link in the paper for downloading the benchmark?
>
> Thanks for the question. We will release the dataset later after the internal review.
>
> ---
>
> > Q3: The paper states: "For a question to pass the screening stage, a minimum correctness of 95% was required from both reviewers." I didn't understand this -- what does "95% correctness" mean for a single question?
>
> For each question, at least 80% of the humans should get it right. Otherwise, we would modify the questions, and new annotators will complete the modified questions. Our marginal threshold is 95% percent. We lower to 80% as we realize that it might be a bit too high.
>
> ---
>
> > Q4: The paper states: "current interleaved image understanding and video understanding models cannot be effectively compared on the same question." What does "interleaved image understanding and video understanding" mean? [I don't know what you mean by "interleaved" here.] And why can't current models be compared on the same question?
>
>
> The term "interleaved" refers to a general input format where images and texts are in the input of the model (the questions) in an interleaved status, meaning there will be multiple images and multiple chunks of text in the question.
>
> An example is
>
> <image-placeholder: a1017.png>If you put these given pieces together, they make which one of these shapes:\nA. <image-placeholder: a1017d.png>\nB. <image-placeholder: a1017a.png>\nC. <image-placeholder: a1017b.png>\nD. <image-placeholder: a1017c.png>\nPlease answer with the option's letter A, B, C, D and explain to me like I am 11 years old.
>
> ---
>
> > Q5: I also am not sure what you mean by "interleaved image-text datasets" and "multi-video interleaving and video-image interleave formats".
>
> The “interleaved image-text” refers to the question being in the format of image and text interleaving.
> The "multi-video interleaving and video-image interleave formats" refers to the format where there will be multiple videos, images and texts interleaving.
>
> An example of multi-video interleaving and video-image interleave format is:
>
> Is the <image-placeholder> in the following two videos <video-placeholder> and <video-placeholder>?
>
> ---
>
> > Q6: I really did not understand the description of the "Human Benchmark". You say that there were 22 participants.
> Then you say "Each annotator was asked to label 2 to 6 concepts". Are the 22 participants acting as labelers, or as participants answering questions from the dataset? What are the human labels used for? You also say "Participants were instructed to skip a question if the question is worded ambiguously or is too complicated to answer in 90 seconds". Are these participants answering questions from the benchmark? What happened to those questions that people skipped? Were they taken out of the benchmark and not used for LLM evaluation? All this needs a much clearer explanation.
> Also, human performance is given in Figure 5. How many humans were tested on each benchmark question? How many benchmark questions were given to each human? Say more about the human study -- how were they presented with the questions? Were there questions given to LLMs that weren't given to humans? Can you give some statistical significance measures to the differences between models and between humans and LLMs?
>
>
> We clarify the confusion as follows. By "Human Benchmark", we mean "Human Baseline". There are a total of 22 participants engaged in this round of evaluation. Each participant is paid for the amount of work they have done. So, the least amount of work is 2 bundles of concept questionnaires and the most amount of work one has done is 6 bundles of concept questionnaires. They are instructed to skip when one question takes more than 90 seconds. We have modified the question and asked other annotators to redo the questions. Each question has 10 answers from different human beings. We would love to further clarify this process.

---

> ### Author Response · Authors · 2024-12-02
> **Response to reviewer 6bzH**
>
> > Q7: Also, Figure 5 is very hard to read. It would be useful if you just picked the five best models, and gave their accuracies compared to humans in a table, and put the rest of the results in an appendix. Also, for the LLM results, which prompt format was being used?
>
>
> Thanks for your suggestion. We will revise Figure 5 as suggested for better clarity. For all the results of MLLMs, we use the default prompt, i.e. the “empty strings” which is the question itself. We did not provide any results by LLMs as we only focus on vision cognitive development and the core cognitive abilities of MLLMs.
>
> ---
>
> > Q8: The paper states that for the questions in the benchmark: "the textual information is highly relevant to but does not overlap the image content". It would help to give an example, or point to one already in paper.
>
> Thanks for raising the question. We will clarify this and revise the paper to be more comprehensive.
>
> ---
>
> > Q9: The paper states: "GPT families show moderate performance, with accuracy scores between 0.4 and 0.6". For these and other results, what is random chance baseline? Why do you call this "moderate" performance?
>
> Thanks for the question. As we have different types of questions on each concept, the exact numbers for random chance baselines will vary across the subsets of the benchmark. But overall, chance performance is around 30% given most of our questions are multiple-choice questions with over four choices. Given that, we have interpreted accuracy between 0.4-0.6 as moderate performance, while accuracy below 30% (e.g. most model’s performance on perspective-taking tasks) is referred to as poor performance. We aim to clarify these terminologies and the precise numbers in the final version of the paper.
>
>
> ---
>
> > Q10: For Table 2, What is the prompt in "empty string" condition Just the question itself plus images? Give an example of an actual prompt, including the question itself. For all the prompt formats listed in Table 2, where does the prompt text go with respect to the question itself?
>
> The "empty string" condition refers to the plain questions with a modality placeholder without other prompts.
>
> We give an example here for better illustration.
>
> Empty string:
>
> Please count the items in the image and answer: Are there more dogs or more animals in the image<image-placeholder: h0001.png>?\nA. Dogs\nB. Animals\nC. The same\nPlease answer with the option's letter A, B, C directly.
>
> Let’s think step by step:
>
> Please count the items in the image and answer: Are there more dogs or more animals in the image<image-placeholder: h0001.png>?\nA. Dogs\nB. Animals\nC. The same\nPlease answer with the option's letter A, B, C directly. Let’s think step by step.
>
> ---
>
> > Q11: Also for Table 2: Are the values given in the table the accuracies over all questions in the dataset?
>
> Yes. All values are accuracies over all questions in the dataset. For instance, 0.52 means that 52 % of the questions are answered correctly by the model.
>
> ---
>
> > Q12: Table 2: "Concept: [concept description]". Give an example of a concept description that would appear here.
>
> Thanks for the question. Please see an  example of the concept description prompt (regarding the concept “Hierarchy”) below:
> Please read the concept explanation then answer the related question. Concept: Hierarchy refers to the cognitive phenomenon that children begin to understand hierarchical relations and be able to organize objects or concepts into structured categories and subcategories, which are supported by the development of mental operations marked by class inclusion and transitivity. Class inclusion refers to the ability to recognize that some classes or groups of objects are subsets of a larger class. For example, a child in the concrete operational stage is able to understand that all roses are flowers, but not all flowers are roses. This concept is essential for one’s systematic and logical organization of conceptual knowledge. Complementarily, transitivity refers to the ability to understand logical sequences and relationships between objects. For instance, if a child knows that Stick A is longer than Stick B, and Stick B is longer than Stick C, they can deduce that Stick A is longer than Stick C. With this ability, they are able to arrange objects within or across categories along quantifiable dimensions. These understandings are crucial when dealing with more complex physical and social situations.
>
> Interestingly, we found that concept description prompting is more effective than standard in-context learning prompting on our benchmark, as shown in table 2, which preliminarily supports that pre-training concept knowledge could be an effective way to enhance MLLM’s performance.

---

> ### Author Response · Authors · 2024-12-02
> **Response to reviewer 6bzH**
>
> > Q13: "Concept explanation...surpasses all the other prompts by at least 6%". I'm confused by this. GPT4o-High for example gets 0.636 with "explanation" prompt and "0.617 with "role assignment prompt." This is less than 3 percent difference. There are other even closer examples in the table.
>
> Thanks for pointing out this problem. We have revised our paper to be more concise.
>
> ---
>
> > Q14: The paper states, for Table 2: "We observe that most prompts are useful on our benchmark, increasing the averaged performance by at least 1%." Increased over what?
>
> Thanks for your question. We mean by increasing compared to the “empty string”, i.e. no prompts, just the question itself.
> We will revise this and improve the comprehensiveness of this paper in the revision.
>
> ---
>
> > Q15: "MMLMs virtually do not understand the answers they produce when tackling complex reasoning tasks". On this you should cite https://arxiv.org/abs/2311.00059
>
> Thanks for raising the question. We will add this citation in the final version.
>
> ---
>
> > Q16: "This study serves as a step toward unraveling the nature of MLMM intelligence and their potential limitations in mirroring human cognitive development." But why should mirroring human cognitive development be a goal?
>
> This is due to the concern that the stage-to-stage transition in which complex abilities are grounded on simple abilities as observed in human cognitive development is critical for better performance. Recent studies have supported the hypothesis that limitations regarding the robustness of current MLLMs on high-level reasoning may be attributed to the lack of core cognitive abilities. It is, therefore, reasonable to hypothesize that MLLMs do need proficiency in core cognitive abilities to fulfill their primary functions effectively, and by assessing whether core cognition has emerged in MLLMs and how their performances on core cognition tasks compared to more complex reasoning tasks on the same domain, our benchmarks are precisely here to probe said proficiency. Further technical analysis of the relevant question can be found in the discussion section of the revised paper.

---

> ### Author Response · Authors · 2024-12-03
> **Response to the additional comments**
>
> We thank the reviewer for their further comments, which are very helpful.
>
> > W1: There have been many updates to the notion of Piaget's notion of developmental stages. See, e.g., https://journals.sagepub.com/doi/full/10.1177/17456916231186611 for an overview. The paper should discuss and cite this literature.
>
> Thanks for the literature provided. The updated view suggesting that there are no clear stages throughout cognitive development, in contrast to Piaget’s notion, is very informative. However, it is worth noting that this notion appears to be more based upon a more mechanistic account of how learning happens rather than how landmark abilities emerge through time, which is what we aim to highlight. In any sense, we will definitely provide further clarifications on how we interpret and incorporate Piaget's theory in further revisions of the paper.
>
> > W2: A "trajectory" implies a dynamic set of transitions over time. What you are measuring in LLMs is not any kind of trajectory, so I recommend using a different term. Perhaps the "hierarchy of capabilities", which in humans is developed over time.
>
> We very much appreciate this suggestion. Our use of the term “trajectory” was aimed to be more descriptive in the sense that simple and complex abilities are causally related in terms of their emergence throughout cognitive development. Given this purpose, using a term that highlights the hierarchical nature of the set of abilities we assessed here is indeed more appropriate. We will thoroughly consider the implications of our analysis and adjust our paper accordingly.

---

### Official Review · Reviewer_rgi9 · 2024-11-02

**Soundness:** 2
**Presentation:** 2
**Contribution:** 2
**Rating:** 5
**Confidence:** 4

**Summary:**

This paper presents CogDevelop2K, a benchmark designed to evaluate cognitive development in multimodal large language models (MLLMs) using stages derived from Piaget’s theory of cognitive development. The authors examine the ability of 46 MLLMs to perform tasks associated with cognitive stages—sensorimotor, preoperational, concrete operational, and formal operational—using visual and textual information. Their results suggest an intriguing yet unexpected “reverse cognitive trajectory,” where MLLMs perform better on complex, abstract tasks than on simpler, foundational ones.

**Strengths:**

The paper’s strength lies in its novel attempt to frame LLM evaluation within Piaget’s developmental theory. This approach is relatively under-explored and introduces an interesting perspective for assessing MLLMs. Specifically, the work highlights cognitive aspects, such as object permanence and intentionality, that MLLMs often struggle with, even though they excel in other domains.

**Weaknesses:**

Despite its ambitions, the paper demonstrates significant limitations. While the authors claim to follow Piaget’s theory, their analysis predominantly relies on measuring sensor-based performance without adequately accounting for the motor and operational aspects critical to Piaget’s framework. This narrow focus restricts the evaluation to text-image understanding, excluding the interactive and embodied dimensions integral to cognitive development, which Piaget considered essential. Though I understand the MLLMs are currently only capable on text and image.

Furthermore, from a practical perspective, the selected benchmarks and developmental concepts provide little insight into the utility of LLMs in real-world applications. Large language models are typically designed for language-based interactions and problem-solving, not to replicate human-like developmental stages. The paper overlooks the fact that LLMs do not need proficiency in these specific developmental tasks to fulfill their primary functions effectively.

Finally, the evaluation results reveal limited takeaways. The findings reiterate known limitations—such as MLLMs’ reliance on pattern recognition over genuine understanding—without offering new pathways for advancing MLLM performance. The observed “reversed developmental trajectory” is an interesting phenomenon, but the analysis falls short of explaining its implications or translating it into practical guidance for AI development.

**Questions:**

The authors might want to consider answering questions and comments raised above.

---

> ### Author Response · Authors · 2024-11-28
> **Response to reviewer rgi9**
>
> We thank the reviewer for recognizing the novelty of our work. We also express our gratitude towards the valuable comments and advice.
>
> > W1: Despite its ambitions, the paper demonstrates significant limitations. While the authors claim to follow Piaget’s theory, their analysis predominantly relies on measuring sensor-based performance without adequately accounting for the motor and operational aspects critical to Piaget’s framework. This narrow focus restricts the evaluation to text-image understanding, excluding the interactive and embodied dimensions integral to cognitive development, which Piaget considered essential. Though I understand the MLLMs are currently only capable on text and image.
>
> First, we take inspiration from Piaget’s theory, yet our goal is to understand the remarkable performance of MLLMs and the mechanism behind it. Although interesting, it would not be necessary to measure the motor and operational aspects if our subject is MLLMs.
>
> Furthermore, as much as we want to measure how embodiment contributes to the performance of MLLMs in Piaget’s framework, the restriction to text and image has largely limited our work to explore the motor and operational aspects critical to Piaget’s framework. Also, we kindly remind the reviewer that such a problem prevailed in the community of vision-language models.
>
> At the same time,  it is critical to note that here our main purpose of adopting Piaget’s framework is to examine the stage-to-stage transitions of cognitive abilities. A key contribution of Piaget’s work is that it provides a comprehensive account of the sequence in which simple cognitive abilities precede more complex abilities on each domain. While some specific details posed in the framework have been questioned (e.g. the age of when a certain ability typically emerges), recent studies generally support the pattern of stage-to-stage transitions between the abilities we investigated here as originally proposed by Piaget.
>
> ---
>
> > W2: Furthermore, from a practical perspective, the selected benchmarks and developmental concepts provide little insight into the utility of LLMs in real-world applications. Large language models are typically designed for language-based interactions and problem-solving, not to replicate human-like developmental stages. The paper overlooks the fact that LLMs do not need proficiency in these specific developmental tasks to fulfill their primary functions effectively.
>
> Thanks for raising the concern. However, we would like to suggest that while it is true that they are not designed to purposefully replicate human-like developmental stages but to solve problems, we leveraged these benchmarks precisely due to the concerns that certain features observed in human cognitive development are critical for better task performance. In particular, our benchmark assesses whether core cognition has emerged in MLLMs and how their performances on core cognition tasks compared to more complex reasoning tasks on the same domain. This is highly meaningful given that recent studies have supported the hypothesis that limitations regarding the robustness of current MLLMs on high-level reasoning may be attributed to the lack of core knowledge. In other words, it is reasonable to hypothesize that MLLMs do need proficiency in core cognitive abilities to fulfill their primary functions effectively, and our developmental tasks are precisely here to probe said proficiency.
>
> ---
>
> > W3: Finally, the evaluation results reveal limited takeaways. The findings reiterate known limitations—such as MLLMs’ reliance on pattern recognition over genuine understanding—without offering new pathways for advancing MLLM performance. The observed “reversed developmental trajectory” is an interesting phenomenon, but the analysis falls short of explaining its implications or translating it into practical guidance for AI development.
>
> We have updated the relevant technical analysis on the revision, which provides a preliminary account of the reason underlying the reversed developmental trajectory we observed and how it implies practical guidance for AI development. In short, we stress that the phenomenon is due to the inability to employ core knowledge in problem-solving. This suggests that further development of MLLM reasoning may benefit from pre-training the correspondence between the conditions of the task and the cognitive domain it is grounded upon, as highlighted by our data showing a unique improvement in performance caused by adding concept explanations in the prompts.

---

> > ### Comment · Reviewer_rgi9 · 2024-12-02
> >
> > Thank the authors for their response, though I do not buy their arguments.
> >
> > While their reference to Piaget’s theory could be considered as a minor wording issue, the authors agree that LLMs are not designed to purposefully replicate human-like developmental stages but to solve problems, and the many argument stay at the hypothesis level without rigorous community-wise recognition. The work did not provide any meaning arguments either.
> >
> > I stand by my initial rating.

---

> ### Author Response · Authors · 2024-12-03
> **Response to additional comments by reviewer rgi9**
>
> We thank the reviewer for their further comments.
>
> > While their reference to Piaget’s theory could be considered as a minor wording issue, the authors agree that LLMs are not designed to purposefully replicate human-like developmental stages but to solve problems
>
> While we acknowledge that LLMs are not designed to purposefully replicate human-like development, we believe that what makes our work valuable is exactly these arguments that are not widely recognized by the community and we aim to provide a brand new and different perspective of looking at MLLMs, inspired by human cognitive developmental theories. Our address to the reviewer’s original comments has already explained, with detailed technical explanations, that we leveraged these benchmarks precisely to assess the argument that essential features of human cognitive development could account for current limitations in MLLMs, hence opening up novel pathways in improving their task performance. Please refer to our response to the original question above and the discussion section of the revised paper for more details.
>
> > and the many arguments stay at the hypothesis level without rigorous community-wise recognition.
>
> The argument we assessed came exactly from the literature [1, 2, 3], which is in turn based on a series of rigorous findings that are widely recognized in developmental psychology and beyond [4, 5, 6, 7, 8].
>
> [1] Brenden M Lake, Tomer D Ullman, Joshua B Tenenbaum, and Samuel J Gershman. Building machines that learn and think like people. Behavioral and brain sciences, 40: e253, 2017.
>
> [2] Melanie Mitchell. On crashing the barrier of meaning in artificial intelligence. AI magazine, 41(2): 86–92, 2020.
>
> [3] Richard Shiffrin and Melanie Mitchell. Probing the psychology of ai models. Proceedings of the National Academy of Sciences, 120(10):e2300963120, 2023.
>
> [4] Elizabeth S Spelke. Core knowledge. American psychologist, 55(11):1233, 2000.
>
> [5] Stanislas Dehaene, Véronique Izard, Pierre Pica, and Elizabeth Spelke. Core knowledge of geometry in an Amazonian indigene group. Science 311(5759): 381-384, 2006.
>
> [6] Elizabeth S Spelke and Katherine D Kinzler. Core knowledge. Developmental science, 10(1):89–96, 2007.
>
> [7] Giorgio, Vallortigara. Core knowledge of object, number, and geometry: A comparative and neural approach. Cognitive neuropsychology 29(1-2): 213-236, 2012.
>
> [8] Renee Baillargeon and Susan Carey. Core cognition and beyond: The acquisition of physical and numerical knowledge. Early childhood development and later outcome, 1, 2012.

---

### Official Review · Reviewer_RFFR · 2024-11-04

**Soundness:** 2
**Presentation:** 3
**Contribution:** 2
**Rating:** 5
**Confidence:** 4

**Summary:**

This paper tries to develop a new dataset that helps answer whether LLMs are stochastic parrots. The authors construct a set of multimodal question-answering tasks grouped by cognitive development stages to evaluate Multimodal LLMs’ core cognitive capabilities. The dataset consists of single and multiple frame questions. The authors tested 46 MLLMs on the proposed benchmark and found that the MLLMs have a reversed cognitive development trajectory.

**Strengths:**

- The dataset consists of a diverse set of cognitive tests that can be beneficial to the community.
- This paper tests the benchmark extensively on different models and prompting techniques.
- The description of the paper is clear and easy to follow.

**Weaknesses:**

- While it is an interesting dataset for evaluating MLLMs, a poor performance on this dataset doesn’t indicate whether LLMs are stochastic parrots. For example, LLMs are not required to have the ability in the sensorimotor stage to answer questions about object relationships.
- MLLMs perform better for questions in the Formal Operation stage may not be a reverse cognitive development trajectory. It can be related to the training data available for training MLLMs, most of them are in the later stage. So, it will be more beneficial to the community to understand why we need to know the cognitive development trajectory for MLLMs.

**Questions:**

- There are some other benchmarks inspired by cognitive reasoning. How does the proposed work relates or differs from those work? For example,
Shu, T., et al. "AGENT: A benchmark for core psychological reasoning. In ICML 2021.

---

> ### Author Response · Authors · 2024-11-29
> **Response to reviewer RFFR**
>
> We thank the reviewer for recognizing the diversity of our cognitive benchmarks, comprehensive testing, and easy-to-follow writing. We also thank the reviewer for their valuable comments and suggestions, which we will address as follows.
>
> > W1: While it is an interesting dataset for evaluating MLLMs, a poor performance on this dataset doesn’t indicate whether LLMs are stochastic parrots. For example, LLMs are not required to have the ability in the sensorimotor stage to answer questions about object relationships.
>
> We do not think that poor performances on certain parts of the benchmark alone indicate that MLLMs are stochastic parrots. In contrast, it is their good performance on high-level tasks co-occurring with poor performance on lower-level tasks that indicates MLLMs’ lack of understanding. In particular, if their performance on high-level tasks are genuinely supported by high-level cognitive abilities, it is unreasonable for them to fail at lower-level tasks which assess basic cognitive abilities that ground said high-level abilities. For example, it would be unlikely for humans who can solve advanced algebra problems to fail at simple addition tasks, given that the latter is foundational to the former.
>
> Regarding the specific example raised by the reviewer, while it is true that answering questions about object relationships in general may not require early-stage cognitive abilities like boundary and spatiality, they could be critical for answering such questions when the tasks involve visually ambiguous scenarios. A lack of such abilities may thus account for the limited robustness of MLLMs’ performance.
>
> ---
>
> > W2: MLLMs perform better for questions in the Formal Operation stage may not be a reverse cognitive development trajectory. It can be related to the training data available for training MLLMs, most of them are in the later stage. So, it will be more beneficial to the community to understand why we need to know the cognitive development trajectory for MLLMs.
>
> It’s for sure that MLLMs’ high performance on Formal Operational stage questions has largely to do with the available training data, which features more information that are directly related to intentionality understanding and tool-using as opposed to e.g. boundary and continuity. However, the interesting question is whether such performances are supported by a genuine proficiency in the underlying cognitive concepts or mainly to do with the reliance on spurious cues. In conjunction with their poor performance at tasks assessing early stage abilities on the same domains, it is likely that their performance is less based on the understanding of high-level concepts. This is why we believe that studying the development trajectory for MLLMs is highly informative because it would help the community to better understand the underlying causes of MLLMs’ current limitations with respect to complex reasoning, particularly in terms of robustness and real-world translations. The cognitive literature of learning has paid a lot of attention to human cognitive development due to the aim of understanding how complex abilities can theoretically be derived from basic and simple abilities. And it has since been suggested that in addition to the increase in computational/representational power, the acquisition of stable, general-applicable complex reasoning abilities may be critically supported by “core” abilities on their respective domains, which are shown to be mastered very early. The findings in our study showing that MLLMs perform highly on complex abilities while failing at a simple abilities on similar domains (e.g. intentionality understanding vs. perspective-taking) raises precisely the concern that the absence of “core abilities” may be responsible for their current limitations.
>
> ---
>
> > Q1: There are some other benchmarks inspired by cognitive reasoning. How does the proposed work relates or differs from those work? For example, Shu, T., et al. "AGENT: A benchmark for core psychological reasoning. In ICML 2021.
>
> We acknowledge the existence of previous social or psychological reasoning benchmarks such as the AGENT [1],  DevBench [2] and ContPhy [3]. However, these benchmarks focus on high-level social reasoning abilities [1], linguistic development [2] and high-level physical reasoning abilities [3] while our benchmarks focus on assessing the relationships between lower-level cognitive abilities and higher-level abilities, from which we generate a critical thesis regarding the absence core cognition in MLLMs and its importance.
>
> [1] Shu, Tianmin, et al. "Agent: A benchmark for core psychological reasoning." International conference on machine learning. PMLR, 2021.
>
> [2] Tan, Alvin, Wei Ming, et al. "DevBench: A multimodal developmental benchmark for language learning." arXiv preprint arXiv:2406.10215. 2024.
>
> [3] Zheng, Zhicheng, et al. "ContPhy: Continuum Physical Concept Learning and Reasoning from Videos." arXiv preprint arXiv:2402.06119. 2024.

---

> > ### Comment · Reviewer_RFFR · 2024-12-03
> >
> > I thank the authors for their reply. However, I respectfully disagree with the argument that performing well in high-level tasks but poor in low-level tasks indicates an MLLM lack of understanding. This argument assumes solving high-level tasks depends on the skills on low-level tasks But this is untrue in many places, e.g., in programming, you can still implement functions and recursions even if you implement addition wrong; in developmental psychology, children can infer the value of goals at ten-month-old but can do counting or have good motor control at a later age. Several research studies have revised Piaget’s theory to address these issues and make it more flexible. It is unclear whether the MLLMs should follow the strict definition of Piaget’s theory as there are factors to determine the sequence of stages. So, while it is a good attempt to map the benchmark to developmental stages, the result still doesn't indicate that MLLMs are stochastic parrots. I'll keep my initial rating.

---

> ### Author Response · Authors · 2024-12-04
> **Response to reviewer RFFR**
>
> Thanks for the further comments. As explained in our paper, our adoption of Piaget’s theory is on a high-level. We do not aim to assess whether MLLMs follow the strict definition of Piaget’s theory, but to emphasize the stage-to-stage transition process in which complex abilities are grounded upon simple abilities throughout cognitive development. We do agree that solving high-level tasks does not depend on the skills on low-level tasks in any case, such as those mentioned by the reviewers. However, it is logically followed that certain high-level abilities require low-level abilities on the same domain, which is what we aim to assess in this paper. A clear example is showcased in Fig.6: MLLMs could solve number conservations while failing at counting the number of coins in the same image featured in the conservation task. If their performance on the conservation task is supported by the understanding of conservation, which logically prescribes rudimentary understanding of numbers, said failure simply could not happen. The explanation could only be that this performance is based on exploitation of spurious correlations, rather than genuine understanding. While we respect the reviewer's decision, we do encourage the reviewer to consider these addresses, which we hope further clarifies the arguments and contributions made in the paper.

---

### Meta-Review · Area_Chair_1EYz · 2024-12-23

**Metareview:**

The paper presents CogDevelop2K, a new benchmark to test visual reasoning capabilities that authors associate to different stages of human cognitive development. The benchmark consists of single-frame and multi-frame questions. The authors find that multi-modal LLMs perform differently than humans. MLLMs tend to do better on complex, more abstract tasks that correspond later stages of human cognitive development than on questions corresponding to earlier stages.

Reviewers raised concerns about using the findings from the benchmark to claim "reverse cognitive development trajectory". Reviewers pointed that these findings are more related to perhaps Moravec's  paradox. All reviewers agreed that more investigation is needed to substantiate the claims made in the paper. Additionally, we encourage the authors to use the reviewer's feedback to include a more thorough discussion of development psychology literature.

**Additional Comments On Reviewer Discussion:**

In addition to points raised in the metareview, there were few other points of discussion.

- using the word "trajectory" implies that there is a notion of time, that LLMs develop along a trajectory in time, but in the study, that wasn't the case. This should be further clarified in the updated manuscript.

- Connections to Piaget's theory are loose without considering motor and operational aspects. While the authors defended their position, reviewer's concerns remained after the discussion.

 - Typos: Multiple reviewers pointed out typos in writing. We recommend the authors to carefully proofread the future manuscript.

---

### Decision · Program_Chairs · 2025-01-22

Reject